# Generating and Visualizing Spatially Disaggregated Synthetic Population Using a Web-Based Geospatial Service

**Jian Liu** [1,2,3], **Xiaosu Ma** [4,*], **Yi Zhu** [5], **Jing Li** [2], **Zong He** [2] **and Sheng Ye** [3]

1. Chongqing University, Chongqing 400044, China; hc048_d@126.com
2. Chongqing Geomatics and Remote Sensing Center, Chongqing 401147, China; leeflylee@163.com (J.L.); hezong@dl023.net (Z.H.)
3. University of Chinese Academy of Sciences (UCAS Chongqing), Chongqing 400714, China; yscc@dl023.net
4. School of Architecture, Southeast University, Nanjing 210096, China
5. School of Public Economics and Administration, Shanghai University of Finance and Economics, Shanghai 200433, China; zhu.yi@mail.shufe.edu.cn
* Correspondence: maxs@seu.edu.cn

**Abstract:** Geographically fine-grained population information is critical for various urban planning and management tasks. This is especially the case for the Chinese cities that are undergoing rapid development and transformation. However, detailed population data are rarely available in comprehensive and timely means. Therefore, appropriate approaches are needed to estimate populations from available data sets in a systematic way to support the continuous demand from urban analytics and planning. Population synthesis approaches such as Iterative Proportional Fitting (IPF) were developed to combine microdata samples with marginal statistics about population characteristics at aggregated spatial levels in order to expand the microdata sample into a complete synthetic population. This paper presents the framework for and the implementation of a geospatial platform for supporting the generation and exploration of spatially detailed urban synthetic populations. The platform provides analytical and visualization tools to support the quick generation of a full urban population with critical attributes based on the latest data available. The case of the synthetic population of Chongqing is used to illustrate the population information and types of visualization that are facilitated.

**Keywords:** distributed modeling; geospatial services; visualization; population synthesis; service-oriented architecture





## 1. Introduction

Geographically detailed urban population distribution is one of the critical information inputs for most urban planning and management tasks. Spatial decision scenarios including the location choices of new public services and facilities, planning of transportation networks, as well as timely disaster management are to a great extent relying on the knowledge of population distribution. As urban management and planning become increasingly fine-grained, there is a growing demand for detailed urban population information that could account for the latest demographic changes and trends.

Traditionally, detailed urban population information is collected via the population census. Because the census is resource and labor demanding, it is launched every five or ten years in most countries. As a result, this census population information usually becomes too outdated for the tasks at hand, in particular for those cities currently undergoing rapid development. In addition, it is often difficult to acquire micro-level census data. In some countries such as the United States or Canada, a 1% sample of the census population is publicly available. However, in more countries, only census information at the aggregated geographical level is readily available. Consequently, many urban practitioners and researchers actively seek alternative approaches to acquire a set of data that can capture the timely distribution of the urban population.

In the field of urban and transportation modeling, population synthesis approaches were developed to combine microdata samples that lacked spatial detail with marginal distribution data about population characteristics at aggregated zonal levels in order to expand the microdata sample into a complete synthetic population. Beckman et al. [1] are among the researchers who first proposed to combine the aggregated control totals from the Census Summary Files with the Public-Use Microdata Sample (PUMS) data to generate a relatively complete realization of synthetic households with important attributes. The aggregated statistics of the urban population are usually published by the government at a more frequent tempo. Therefore, they can be used for the timely updating of the urban population. However, at the moment, the usage of synthetic population is mostly restricted in the field of urban modeling and analytics. The result has not been made accessible for the public and private sectors in need of reliable population data in their working routines.

In parallel, the advances in GIS technology and the explosion of spatially detailed data have made geospatial platforms growingly popular as a web-based entry point to geographic content. Due to the advantage of temporal accessibility and resource cataloging, the spatial data infrastructure could attract a higher level of usage [2]. However, the majority of geoportals or spatial data infrastructures (SDIs) have put emphases on geographic information visualization and sharing by providing online mapping with a growing list of additional functions such as direct access to raw data in multiple formats, complete metadata, and online visualization tools and commenting mechanisms [3–5]. Few examples explicitly integrate the population synthesis process and geo-visualization to support the exploration and refining of the urban population at a continuous rate.

Therefore, in this study, we prototyped a Geospatial platform that loosely couple the off-the-shelf open-source applications with proprietary tools to support the synthesis, visualization, and exploration of urban population data that are not only relevant to urban modeling and analytics but also are accessible to urban planners and managers. This paper explains the framework of the geospatial platform for population synthesis, using its implementation in Chongqing, China as an example to illustrate its functionalities.

## 2. Previous Research

A review of the literature indicates an evident deficit in the development of a geospatial platform for the generation and exploration of synthetic population. However, there is considerable research on the related topics. Therefore, in this session, we summarize the related work in the following two research areas: (1) a geospatial platform for urban analytics and modeling; (2) population synthesis approaches.

### 2.1. Geospatial Platform for Urban Analytics

With the development of GIS technology and the rapid growth of geospatial data, geospatial web services such as spatial data infrastructure (SDI) and geospatial portals have become an important branch of web applications. The concept of geospatial web services was coined to describe the revolutionary transition of geospatial services from offline to online. Central to the geospatial web services is the idea of accessing catalog services, geospatial data, and geoprocessing services in a distributed environment over the network. The web-based geospatial service standards developed by OpenGIS Consortium, Inc. (OGC) such as Web Map Service (WMS), Web Feature Service (WFS), and Gazetteer have greatly facilitated the development of geospatial web services in the field of urban planning and management.

To provide real-time forecasting to support coastal research and decision making, Agrawal et al. [6] proposed a geospatial infrastructure (GCI) with developed middleware for model integration/analysis, data mining, and cross-application integration. Hong et al. [7] proposed City Building Energy Saver (CityBES), a web-based platform to simulate the energy performance of a city's building stock, which combines energy modeling and analysis with web-based GIS processing and visualization. Ferreira et al. [4] developed a Geoportal that couple the geospatial web service with the Land Use, Trans-

portation and Environment (LUTE) modeling, in order to support visualization and exchange of data relevant to the complex urban modeling and to accelerate collaboration and model development among research groups working on various components of the LUTE models. The visualization and data exchange capabilities of the Geoportal developed by Ferreira et al. were further consolidated by Zhu et al. [5] by adding an ArcGIS map server and a number of Ajax/Drupal based front end applications.

Monlina and Bayarri [8] introduced the Andean Information System for Disaster Prevention and Relief (SIAPAD), which was developed to integrate and share multinational spatial data about disasters. They used a knowledge-based system on top of the GEORiesgo, a thematic Spatial Data Infrastructure (SDI), to help users find the needed information. Granell et al. [9] investigated the opportunities and challenges of geospatial information infrastructure with standards-based geospatial services, technologies, and data models in health applications. They found the aggregated nature of most health-related data becomes a challenge limiting the range of analysis and processing in health applications.

It is generally clear that geospatial web services have brought and will bring many positive impacts on urban planning and management. Geospatial services like WFS and WCS provide unique opportunities to retrieve geospatial data located at different places and developed for different purposes, which facilitates data transferring and processing [10]. Geospatial services make mapping spatial data easy. For most urban agencies, using geospatial services means a reduction of data processing and management burden. A federated repository of datasets is made possible by dynamically publishing and retrieving the latest data through services like GeoSynchronization [11] and GeoRSS feed. Geoprocessing services also enable urban planning and management agencies to reduce investment in the modules associated with a proprietary GIS package that provides the same functions.

However, accompanying these benefits, some issues also emerge. Urban data analysis, plan making, and urban management routine tasks have not received sufficient support from most existing geospatial web services. Many urban governments have not established a geospatial cyberinfrastructure to publish and share urban spatial information, not to mention the detailed urban demographic information.

*2.2. Population Synthesis Approaches*

Population synthesis is a technique that draws on the information from a set of sample data and aggregated distributions of demographic attributes to generate a synthetic population. The population synthesis has been used widely in urban transportation and land use modeling, as well as in the agent-based urban simulation systems in particular, such as UrbanSim [12], ILUTE [13], and SimMobility [14].

The Iterative Proportional Fitting (IPF) procedure, initially proposed by Deming and Stephan [15], is a popular population synthesis technique in the field of transportation demand and urban modeling. Beckmen et al. [1] applied the IPF procedure to generate a synthetic population for activity-based models. After Beckmen et al. [1], many efforts have been made to improve the IPF procedure to create better realizations of the population of households and individuals. Arentze et al. [16] developed a two-step fitting procedure, which first converted the known marginal distributions of individual attributes to marginal distribution of household attributes, and then used the household marginal distributions for the IPF procedure. Similarly, Zhu et al. [17] presented a two-step IPF approach to fit both household and individual-level constraints simultaneously. Guo and Bhat [18] proposed a generic IPF algorithm to deal with the zero-cell problem and to improve the scalability and transferability of the approach. Auld and Mohammadian [19] also presented an improved version of the basic IPF procedure by accounting for the multiple-level marginal controls. Recently, Otani et al. [20] found the modifiable attribute cell problem (MACP) arising from categorizing attributes from contiguous attributes and proposed to identify the best-organized cell set using a five-step solution.

Aside from the IPF approach, researchers have experimented with other techniques to resolve the population synthesis problem. Ye et al. [21] proposed a heuristic approach to iteratively adjusting weights of households of a specific type until the multi-way constraints are matched. Barthelemy and Toint [22] developed a synthesis method working without samples based on a hierarchical three-step approach including entropy maximization and Tabu search, and ad hoc rules. A simulation-based approach was developed by Farooq [23] to address the data limitation issue. Gibbs sampling was employed to produce the joint distribution of individuals' attributes based on a series of conditional distributions estimated based on the limited observed data. Other sample-based approaches such as the Bayesian network [24] or the Hidden Markov model [25] have emerged.

In most previous research, the level of the spatial unit of population synthesis was typically at the traffic analysis zone or other zonal levels. This satisfies the requirement of transportation modeling which has a relatively loose demand for spatial resolution. However, many other urban planning and management tasks, such as community planning, school district delineation, and public service location choice, demand more spatially detailed demographic information.

Because the Iterative Proportional Fitting (IPF) procedure is a mature methodology theoretically and computationally, the geospatial platform in this study employs the two-step IPF approach proposed by Zhu et al. [17], with an allocation model assigning synthetic households to the residential projects.

### 2.3. Contributions

In this study, we proposed a population synthesis geospatial platform, which would make two major contributions to the existing literature. First, urban planning and management tasks require population data at a fine-grained spatial resolution. However, many existing research approaches generate synthetic populations at the zonal level, where aggregated population statistics are available. In this study, the conventional population synthesis process was consolidated by including a population allocation model, which is intended to assign synthetic households to more spatially detailed locations. Second, by encapsulating population synthesis as a web-based geospatial service, we enable the processing, visualization, and exploration of synthetic populations by different parties. The case of Chongqing demonstrates the potential of allowing urban planners and managers to access more detailed and updated demographic information via a geospatial cyberinfrastructure.

## 3. Population Synthesis Methodology

### 3.1. Procedure of Synthetic Population Generation

The major objective of the population synthesis process is to re-populate the sample households and individuals which minimizes the discrepancy between the attribute distribution and those observed in various marginal distributions recorded in official statistics. Given the population sample data, marginal control statistics of demographic attributes at the zonal level, and the residential location data are readily available, the realization a synthetic population can be generated through the proposed procedure as follows (Figure 1):

1. Estimating the number of residential units in each residential area: a regression-based model is built to estimate the number of units in those residential areas with missing capacity information.
2. Generating marginal distribution constraints for demographic attributes from aggregated population statistics: the marginal distribution constraints of household and individual attributes such as age group, gender, and household size were extracted from zonal population statistics available at official sources such as statistic yearbook.
3. Using Iterative Proportional Fitting (IPF) to fit the sampling weights for household and individual samples: the sampling weights of population samples are calculated by iteratively matching the attribute distribution of the current synthetic population

to corresponding marginal distribution constraints until the convergence criteria are met.

4.  Re-sampling the population based on the sampling weights generated at step 3.
5.  Assigning the synthetic population to appropriate residential areas: the synthetic population generated is then assigned to residential areas using a capacity-constrained allocation model.

In what follows, we provide a detailed description of the two-step IPF method and the capacity-constrained allocation model.

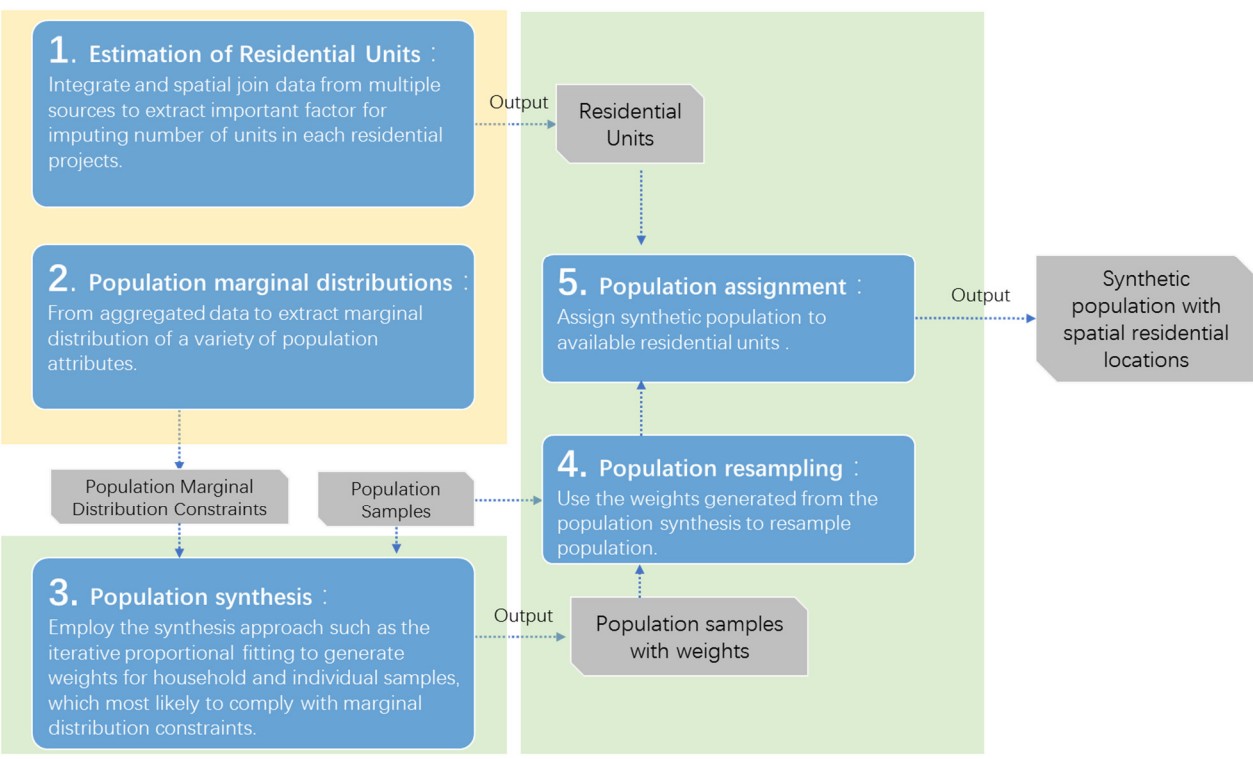

**Figure 1.** The analytical procedure of population synthesis.

### 3.2. Population Synthesis Method

The population synthesis approach proposed by Zhu and Ferreira [26] is employed in this study to estimate the joint distribution of multidimensional attributes, which combine the correlation information from the microdata sample and the reliable marginal distribution constraints obtained from independent sources. The typical objective function for a population synthesis problem could be formulated as Equation (1), originally proposed by Deming and Stephan [15]:

$$\sum_i (p_i - \hat{\pi}_i)^2 / p_i$$
$$s.t. \begin{cases} \sum_{i \in j} \hat{\pi}_i = r_j \ for \ j = 1, 2, 3, \ldots, J \end{cases} \tag{1}$$

where *i* represents an n-way cell within an n-way matrix representing the joint distribution of population attributes; *n* is the number of constraint attributes; $p_i$ denotes to the sample proportion of cell *i* and $\hat{\pi}_i$ is the estimated cell proportion. *j* is the index for attribute categories and *J* is the total number of categories of all constraint attributes. Marginal distributions $r_j$ are extracted from reliable sources for the estimation of $\hat{\pi}_i$. Deming and Stephan proposed the standard IPF algorithm to estimate $\hat{\pi}_i$.

Zhu and Ferreira [26] expanded the standard algorithm in order to satisfy multiple levels of marginal constraints (household and individual level) simultaneously. They proposed to solve a nonlinear equation at each iteration when the expansion factors of households are estimated by the IPF procedure to meet the personal level constraints. In that case, Equation (1) was rewritten as Equation (2) to deal with constraints from both household and individual level statistics:

$$\sum_i (p_i - \hat{\pi}_i)^2 / p_i$$
$$s.t. \begin{cases} \sum_{i \in j} \hat{\pi}_i R_{ij} = r_j \; for \; j = 1, 2, 3, \ldots, J \end{cases} \tag{2}$$

when $R_{ij}$ is used to convert the cell values for households to corresponding individual-level estimates. For example, age as an individual attribute can be converted to multiple separate household attributes (e.g., number of households in each age group). In the iterative procedure, the expansion factor for category $j$ of iteration $t$ can be updated by solving:

$$\sum \hat{\pi}_i^{t-1} \left( \alpha_j^{R_j} \right)^t = r_j \tag{3}$$

The expansion factor $\alpha_j^{R_j}$ plays a critical role in adjusting the household sampling values when the estimated distribution is inconsistent with the targeted distribution revealed by constraints.

### 3.3. Capacity-Constrained Allocation Model

The synthetic population is usually generated at the zonal level constrained by the spatial resolution of the marginal distributions of attributes. However, it is often necessary to further allocate synthetic households to realistic disaggregated locations, for example, buildings or residential projects. In the population synthesis, we adopted a multinomial logit model (MNL) to estimate the likelihood of various types of residential properties to be associated with a given household based on their characteristics. Specifically, the housing properties are categorized by their property types, finished years, and transaction values.

Within the multinomial logit model framework, the probability that a household $i$ lives in a residential building type $j$ conditional on household characteristics $\{x_i\}$ can be formulated as:

$$P(j|i) = \frac{e^{(\alpha_j + \beta_j x_i)}}{\sum_{k \in J} e^{(\alpha_k + \beta_k x_i)}} \tag{4}$$

where $\alpha$ is the intercept, $\beta$ is a vector of coefficients corresponding to variables related to the household. $J$ is the collection of residential building types. Traditionally, the coefficients are estimated by maximizing the log-likelihood function. Using the estimated parameters and Equation (4), the MNL model can predict a set of probabilities representing the chances that a household is in different types of residential buildings. This model however does not resolve the allocation problem because the proportion of selecting residential buildings needs to be constrained by the capacity.

To assign the synthetic population to residential projects that have limited capacities, we employed the capacity-constrained allocation algorithm proposed by De Palma et al. [27]. The detailed allocation steps are described as follows:

1. Given the aggregated population statistics are available at the zonal level, for zone $d$, check if the aggregated number of housing units is greater than the aggregated synthetic households.
2. Iteration 0: set household allocation ratio for all residential areas to 1. For each residential area $j$ within zone $d$, compare the relationship between supply $S_j$ and demand $D_j$, where $S_j$ is the number of housing units in the residential area $j$, and $D_j$

is the expected demand represented by the aggregated probabilities of all synthetic households living in the residential area *j*.

3.　Identify the subset *C(0)* of alternative residential areas that satisfy $(S_j < D_j)$.

4.　Compute the allocation ratios of household *i* using the following equation:

$$\omega^i(0) = \frac{1 - \sum_{j \in C(0)} \frac{S_j}{D_j} p(j|i)}{1 - \sum_{k \notin C(0)} p(k|i)} \tag{5}$$

5.　Compute the alternative-specific allocation ratio for each residential project *j*:

$$\omega_j(0) = \frac{1 - \sum_i \omega^i(0) p(j|i)}{1 - \sum_j p(j|i)} \tag{6}$$

6.　Update iteration: $l \to l + 1$; Update the constrained choice subset $C(l + 1)$:

$$C(l+1) = \{ j \in J \mid S_j < \omega_j(l) D_j \} \tag{7}$$

7.　Repeat step 4 and 5 to calculate $\omega^i(l + 1)$ and $\omega_j(l + 1)$;

8.　Stop at iteration $l + 1$ if $C(l + 1) = C(l)$; else go to step 6;

9.　Using adjusted MNL Equation (8) to calculate the final allocation probabilities of residential project *j* for household *i*:

$$P(j|i) = \frac{e^{(\alpha_j + \beta_j x_i + \ln(\pi_{i,j}))}}{\sum_{k \in J} e^{(\alpha_k + \beta_k x_i + \pi_{i,k})}} \tag{8}$$

where $\pi_{i,j}$ is the utility correction factor for household *i* and residential project *j*, which has the form of

$$\pi_{i,k} = \begin{cases} \frac{S_j}{D_j} \ if \ S_j < D_j \\ \omega^i(\text{end iteration}) \ if \ S_j \geq D_j \end{cases} \tag{9}$$

The imbalance between the demand and supply is resolved by iteratively estimating individual and aggregate allocation ratios $\omega^i$ and $\omega_j$. De Palma et al. [27] pointed out that this algorithm could converge with the capacity constraints.

## 4. Framework of the Population Synthesis Platform

The population synthesis and allocation procedure is encapsulated in a web-based geospatial platform to provide processing, generation, and exploration of the synthetic population for urban planning and management. The framework of the platform centers on an R-based analytical module, a federated database server, and a web-based visualization application with a service-oriented architecture (SOA) as diagrammed below (Figure 2). The platform is mostly built upon open-source applications or tools, such as Apache, PostgresSQL, R, python, with some integration of proprietary tools such as Mapbox and ArcGIS. On the system architecture side, there are four layers: database layer, data analytical layer, middleware, and presentation layer. The database layer stores original and model-derived data. The middleware layer contains the map server (MapServer), web server (Apache), and other application services to support data processing and visualization at the presentation layer. The presentation layer contains portals to provide user interfaces for accessing, viewing, querying, and interacting with the synthetic population data and models. The data analytic layer includes a series of modular applications for population synthesis.

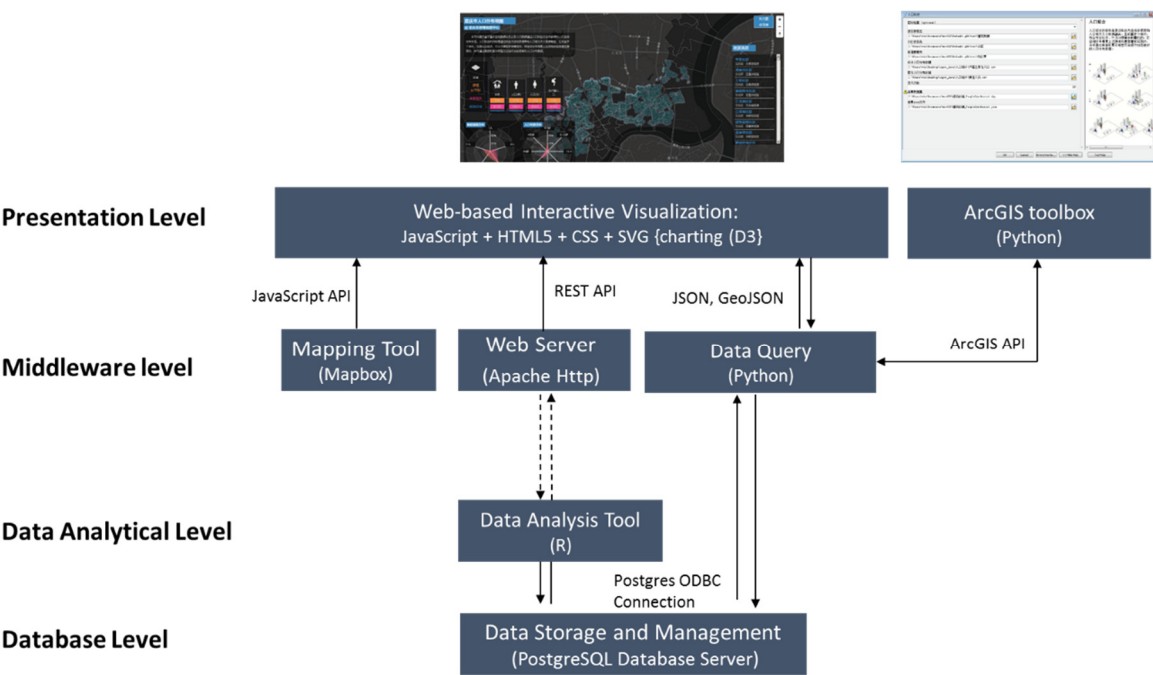

**Figure 2.** System framework of the population synthesis platform.

The system adopts PostgreSQL with its geospatial extension PostGIS to store relational datasets and spatial datasets respectively. It also stores the results of population synthesis and exchanges data with analytical models, application servers via SQL queries across Open Database Connectivity (ODBC) and web services. In order to maximally satisfy the demand from urban planners and managers, we developed two types of tools for interactive data processing and visualization: (1) a JavaScript-based web interface linked to MapServer; and (2) a toolbox built on the top of ArcGIS software using ArcGIS API. Mapserver provides flexible web-based mapping services, while ArcGIS software has been widely used by urban planning professionals and is interoperable with common desktop GIS packages. A user does not need to own or learn each of the complex and/or proprietary software packages since the standardized original data and final model results can be explored and shared through the visualization applications. There are two ways to initiate the population synthesis process: through the web-based platform or through a python coded toolbox embedded in the ArcGIS software. As a result, the user can generate and visualize the resulted synthetic population on the web or on the ArcGIS desktop.

The final output dataset of the population synthesis module is a set of synthetic populations with demographic attributes and residential locations. The core of the population synthesis analytic procedure is developed in R, with the help of the library "Remake" to organize and manage the workflow of the procedure by specifying the inputs, outputs, dependencies, targets, and rules for each analytical and processing step. "Remake" is a "make-like" configuration package to declare an analytical pipeline for R. "Remake" is used to not only minimize the revising effort when some parts of the procedure need to be revised and updated, but also to create reproducible research and analytical workflows for complicated tasks.

In summary, the system is intended to be a Software as a Service (SASS) geospatial platform, which allows users to generate and access synthetic populations through the internet or via desktop GIS packages. The first version of the system was developed in 2018 and has been used to generate and visualize synthetic populations for a number of cities in China.

## 5. Case Study

This section introduces the implementation of the geospatial platform for generating and exploring synthetic population for the Chongqing City of China. The synthetic households were generated and allocated to the residential projects with the assistance of a capacity estimation model to estimate the number of residential units within each residential project, as well as a capacity-constrained allocation model fulfilling the spatial allocation.

### 5.1. Residential Capacity Estimation Model

To generate a synthetic population with fine-grained spatial resolution, it is necessary to have a comprehensive housing stock dataset with capacity information. However, such a dataset is usually not readily available for most Chinese cities. Therefore, we combine the relevant data from multiple sources including the types of land use, and the information from some residential projects from real-estate agencies such as Lianjia to form a relatively complete housing dataset. The datasets were spatially joined, and the housing data were enriched with attributes such as land-use type, designated floor area ratio, types of residential development allowed, and floors, finished years, etc.

Only a subset of the residential projects has information on capacity (i.e., number of housing units in each residential project). Therefore, a linear regression-based estimation model has to be estimated to impute the number of units for the remained residential projects within the combined housing dataset. The dependent variable is the number of units of residential projects, taking the logarithm form. The explanatory variables include the logarithm of total residential floor area, which is calculated from the area of the building footprint and the floors of the buildings, the floor area ratio (FAR) of each residential project, the proportion of Type I residential area, as well as the proportion of Type II residential area. Type I residential area is classified as low-density residential areas with buildings typically lower than three floors. Type II residential area is classified as mid- or high-density residential areas. In contrast, Type III residential area, which is the reference land-use type in the model, is a mixed residential and industrial area. Total residential floor area is calculated by multiplying building footprint areas with the number of floors in each building and then aggregated with the entire residential project. Note that Type I and Type II residential areas are defined in the "Code for Classification of Urban Land Use and Planning Standards of Development Land" (GB50137-2011), China.

As expected, the total residential floor area and the types of residential land use are significantly correlated with the capacity as listed in Table 1. Designated FAR, however, is not significant, because the development intensity has somewhat been captured by the total residential floor area. The regression model has an adjusted R-square value of 0.9745, indicating that the model has a high degree of fitting.

**Table 1.** The estimation results of the number of housing units.

| Explanatory Variable | Coefficient | Std. Error | |
| --- | --- | --- | --- |
| The log of total residential floor area | 0.681 | 0.059 | *** |
| The square of total residential floor area | $6.86 \times 10^{-12}$ | $5.97 \times 10^{-12}$ | |
| The proportion of Type I residential area | −2.04 | 0.708 | *** |
| The proportion of Type II residential area | −1.00 | 0.598 | * |
| Designated Floor Area Ratio (FAR) | −0.025 | 0.024 | |
| Adjusted R-squared | 0.974 | | |
| F-statistic | 1423 on 5 and 186 DF, $p$-value: $< 2.2 \times 10^{-16}$ | | |

Note: Significance code: * $p < 0.1$; *** $p < 0.01$.

### 5.2. Population Sample Preparation and Synthesis

The marginal distribution constraints of population statistics are collected from two sources: the aggregated population statistics at the sub-district level (also known as

"Jiedao") from the local government, and the aggregated population statistics at the district level from a 1% sampling survey conducted in 2015. In Chinese cities, the sub-district is a lower-level administrative division of a city district but is higher than the neighborhood level. The main attributes of the aggregated population statistics are listed in Table 2, including gender, age group, and household size, household structure, and age and floor of the residential buildings. These aggregated statistics of population attributes were served as constraints in the process of population synthesis.

**Table 2.** Marginal distribution constraints of population attributes.

| Attribute | Categorical Value | Spatial Aggregation Level | Sources |
|---|---|---|---|
| Gender | Male, Female | Sub-district | Local government |
| Age | Age between 0–2, 3–4, 6–12, 13–15, 16–18, 19–23, 24–30, 31–55, 56–59, 60–64, 65+ years | Sub-district | Local government |
| Registration status | Local, non-local | Sub-district | Local government |
| Household size | 1, 2, 3, 4, 5+ | district | 1% population survey of 2015 |
| Household structure | 1, 2, 3, 4 generations | district | 1% population survey of 2015 |
| Floor of residential buildings | Flat, 2–3, 4–6, 7–9, 10+ | district | 1% population survey of 2015 |
| Age of residential buildings | <1989, 1990–2000, 2001–2009, 2010+ | district | 1% population survey of 2015 |

Note that the interval of age distribution is classified in accordance with typical socio-economic status, for example, primary school, middle school, college, retired, etc.

The micro-level population sample data are from the China Family Panel Studies (CFPS), which is an annual longitudinal survey of Chinese communities, families, and individuals launched in 2010 by the Institute of Social Science Survey (ISSS) of Peking University, China. The population samples include 824 households and 2160 individuals.

The general IPF method fits the joint distribution of selected household and individual attributes simultaneously for each spatial aggregation level at which reliable marginal totals are available. See Section 3.2 and Zhu and Ferreira (2014) [26] for more details on the IPF approach. Then, the synthetic population is generated by resampling samples based on the fitted weights.

*5.3. Allocation to More Disaggregated Spatial Level*

The synthetic population generated satisfies constraints at the aggregated geographical level. However, urban planning and management tasks usually require the spatial heterogeneity of agents' characteristics captured at the fine-grained spatial level. Using the housing capacity and attributes extracted in Section 5.2, it is possible to assign the generated synthetic population to the residential projects and even residential buildings by matching the overlapping attributes such as floor and age of residential buildings.

Table 3 shows the residential allocation model estimated using the CFPS, after joining the home location of the CFPS samples to the residential projects. The residential projects are classified into nine types, based on the levels of the average transaction price and the time period built. The purpose of using a residential area classification model instead of a spatial location choice model is to approximately match types of residence with the types of households under the circumstance that the micro-level data is insufficient. A multinomial logit model was used with the low-price residential projects built before 1980 as the reference category.

**Table 3.** Estimation results of residential allocation model.

| | Low Price Level (<25%) | | Medium Price Level (25%–75%) | | | High Price Level (>75%) | | |
|---|---|---|---|---|---|---|---|---|
| | 1980–2000 | >2000 | <1980 | 1980–2000 | >2000 | <1980 | 1980–2000 | >2000 |
| Intercept | −0.859 *** | −1.96 *** | −0.259 | −0.297 | −2.237 *** | 0.701 ** | −0.723 ** | −2.738 *** |
| | (0.233) | (0.252) | (0.248) | (0.224) | (0.257) | (0.232) | (0.246) | (0.321) |
| Household structure (>2 generations) | −0.327 * | −0.402 * | −0.425 * | −0.355 * | −0.297 | −0.053 | −0.203 | −0.164 |
| | (0.201) | (0.217) | (0.234) | (0.206) | (0.219) | (0.220) | (0.230) | (0.259) |
| Household Size | 0.448 *** | 0.709 *** | 0.152 * | 0.387 *** | 0.585 *** | −0.031 | 0.195 ** | 0.380 *** |
| | (0.061) | (0.063) | (0.066) | (0.059) | (0.064) | (0.063) | (0.064) | (0.077) |
| Low household income | −0.280 * | −0.502 *** | 0.033 | −0.267 * | −0.091 | 0.090 | −0.058 | −0.117 |
| | (0.155) | (0.163) | (0.168) | (0.150) | (0.166) | (0.159) | (0.164) | (0.200) |
| High household income | 0.072 | 0.230 | 0.545 * | 0.561 * | 1.013 *** | 0.546 * | 0.918 ** | 0.865 ** |
| | (0.286) | (0.294) | (0.296) | (0.272) | (0.287) | (0.284) | (0.283) | (0.326) |
| Number of workers | 0.241 *** | 0.129 | 0.111 | 0.224 ** | 0.221 * | 0.137 | 0.232 ** | 0.291 ** |
| | (0.088) | (0.092) | (0.096) | (0.086) | (0.092) | (0.092) | (0.093) | (0.109) |
| Local households | 0.783 ** | 0.671 *** | 0.628 *** | 0.924 *** | 1.034 *** | 0.475 *** | 1.019 *** | 1.314 *** |
| | (0.133) | (0.141) | (0.142) | (0.128) | (0.145) | (0.134) | (0.142) | (0.189) |
| Having children | 1.067 ** | 0.341 | 0.825 * | 0.96 * | 0.461 | 0.681 | 1.013 * | 0.849 * |
| | (0.407) | (0.436) | (0.430) | (0.403) | (0.435) | (0.423) | (0.422) | (0.482) |

Log-Likelihood: −12877
McFadden $R^2$: 0.130
Likelihood ratio test: chi square = 941.31 (*p*-value = < 2.22 × 10$^{-16}$)

Note: The numbers in the parenthesis are standard errors of coefficients; Significance code: * *p* < 0.1; ** *p* < 0.05; *** *p* < 0.01.

According to the estimation result, households with more members are less likely to stay in old residential projects. Similarly, households with more working members are also less likely to stay in old residential projects. Household income has a significant and positive correlation with the price level of the housing project. In comparison with local households, migrant households are more likely to live in the old and cheap housing projects. The model has a McFadden R2 of 0.130. With the estimated capacity of residential projects and the multinomial logit model that evaluates the likelihood of a household living in a type of residential project, we can then use the capacity-constrained allocation model described in Section 3.3 to allocate the synthetic population to residential projects in a reasonable way.

### 5.4. Validation of Synthesis Population Results

In order to verify the performance of population synthesis, we compared the marginal totals with the aggregations from five synthetic population realizations. Mean absolute percentage error (MAPE) is used to measure the divergence between the distributions of attributes in the synthetic population realizations and those from the aggregated statistics. The final MAPEs reported in Figure 3 are calculated by taking the average of mean MAPE at the sub-district level for five synthetic population realizations generated by the platform.

In general, the synthetic population has better goodness of fit for household-level constraints such as household size and household generations than the individual level constraints. This is because, in the fitting algorithm, the individual-level constraints have to be converted into the household-level constraints, which might introduce errors. The goodness of fit of the synthetic population realizations is also related to the quality of the population samples. In the sample data, the non-local residents are relatively under-represented, which leads to greater MAPE values for both male and female non-local residents than those for the locals. In addition, the sample population perhaps missed out some of the children of age 0 to 5, and college students of age 19–23, which makes the MAPE for these two age groups much higher than those for other age groups. Overall, the validation result suggests the synthetic population realizations generated by the platform have a relatively satisfactory fitting for the aggregated population statistics from the government. This also demonstrates the great potential of the platform to generate updated and spatially detailed synthetic population realizations that are consistent with the government statistics.

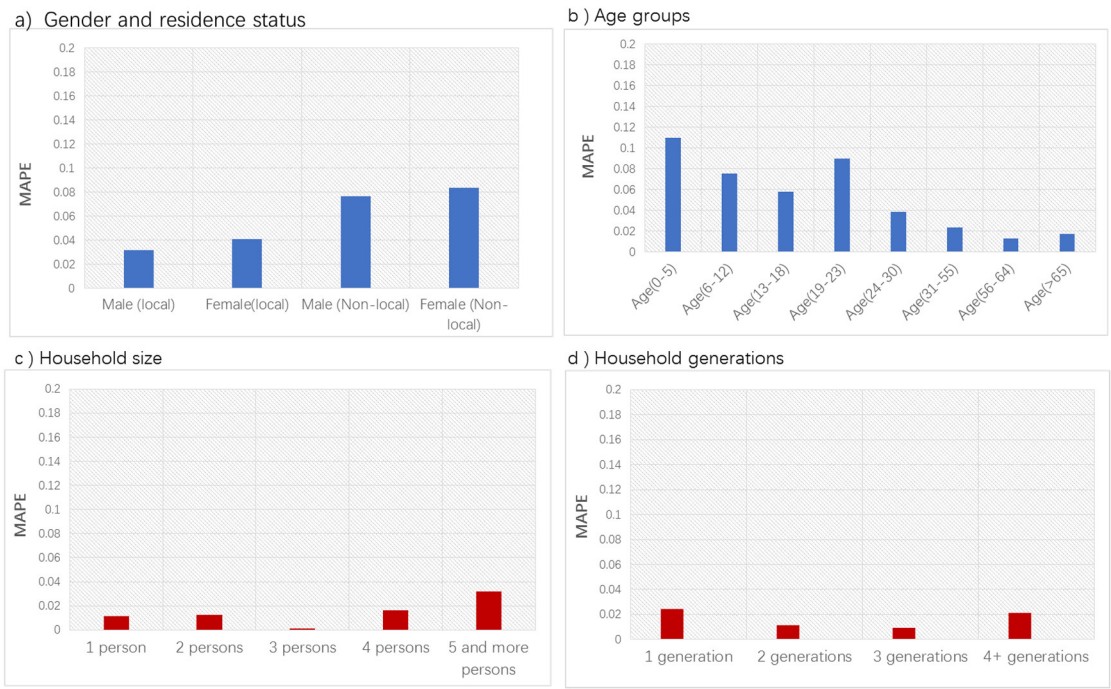

**Figure 3.** Goodness-of-fit results for population generations.

### 5.5. Web-Based Visualization of the Synthetic Population

The web-based visualization application aims to provide a rich exploration and interaction experience for the users. It was developed by JavaScript and employed Mapbox to provide the base map service and rendering of spatial objects. The charting and tabulating were generated with the help of the D3.js library, a JavaScript library for producing interactive data visualizations in web browsers. The features on the map and the related attributes in the table and chart are all visually interlinked.

Figure 4 shows a screenshot of the web-based visualization application. On the log-in page (Figure 4a), the user can first select a district of Chongqing. Then, the population synthesis task could be established, or the existing population realizations could be retrieved (Figure 4b). The visualization in Figure 4c shows a 3D plot of the buildings and the heatmap of the synthetic residential population generated in the analytical module. The map allows users to explore the population distribution in 2D or 3D, with or without a heatmap. It also allows the selection by clicking on a neighborhood from the list on the right or drawing a rectangular area using a mouse. The aggregated statistics and distributions of population attributes of the selected buildings, neighborhoods, or mouse drawn rectangular areas are displayed in columns and radar charts labeled as 3 and 4. This provides a chance to comprehend and compare the spatial differentiability of population distributions at different spatial levels.

To improve the usability of the application, additional interactive effects are added. When one moves the mouse across the map, the corresponding building on the map will be highlighted in red and the relevant data in the charts will also be highlighted. A mouse click on the selected building will pop up a tooltip displaying the name and the attributes of the clicked building.

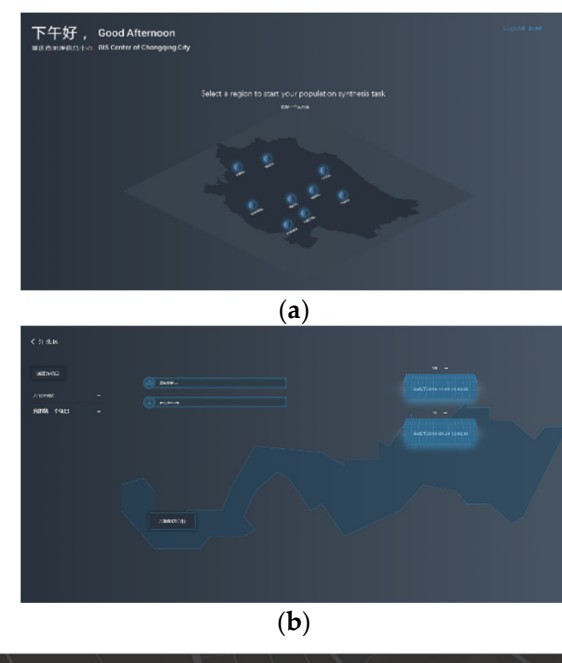

(**a**)

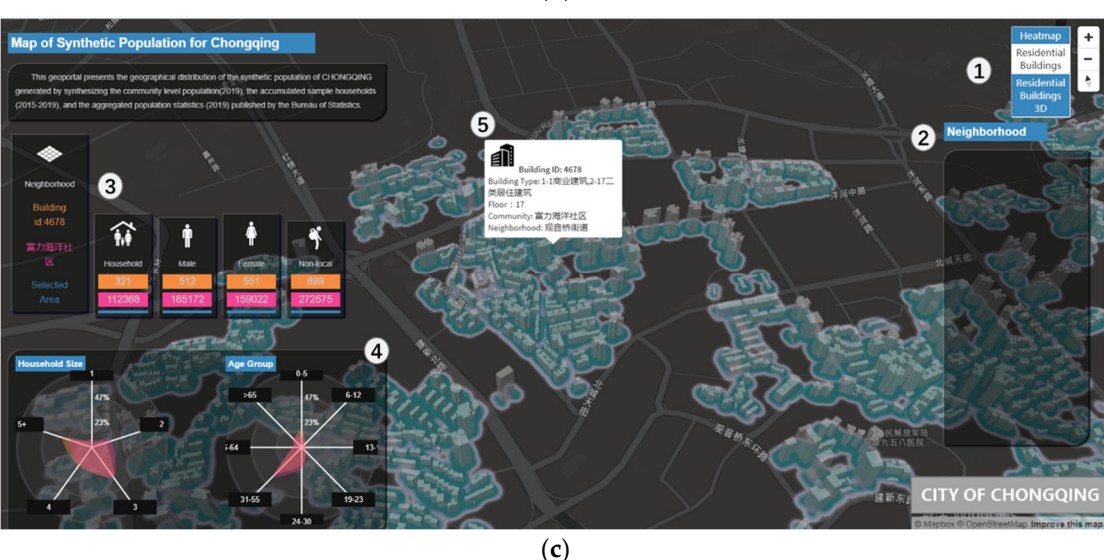

(**c**)

**Figure 4.** Screenshot of the web-based visualization application for a synthetic population in Chongqing. (**a**) Log-in web page; (**b**) Build and manage population synthesis tasks; (**c**) View synthetic population results. (1) Select a visualization mode (2D, 3D, with or without a heatmap); (2) select a neighborhood to zoom-in; (3) count the population statistics (households, individuals, non-locals, males, females) for the selected building, the neighborhood where the building is located, and the area selected by a mouse drawn rectangle; (4) radar charts shows the distribution of age and household size of the population within three levels of spatial areas; (5) click building shows the name and attributes of the building.

### 5.6. Service Transferability

To test the transferability of the platform for a different context, we applied the population synthesis geospatial portal to generate the synthetic population for another city of China, Suzhou. On the basis of the existing modules for Chongqing, we only needed to (1) prepare the sample population data, marginal distribution constraints, and residential areas for the context of Suzhou; (2) set-up the neighborhood and building vectors of Suzhou in the PostgreSQL/PostGIS database and change the mapping environment settings for the visualization application. Figure 5 shows the synthetic population results for the city of Suzhou.

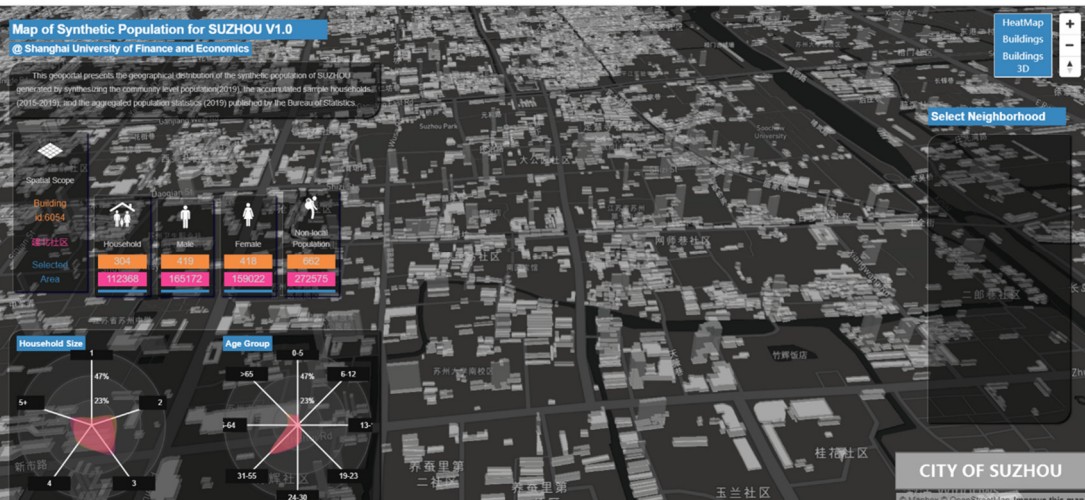

**Figure 5.** Screenshot of the web-based visualization application for a synthetic population in Suzhou.

Both the population synthesis routine and the visualization applications were able to function for the case of Suzhou after a short-time effort. However, the system encountered a few issues in the procedure. Different population datasets introduce new attributes that challenged the old database structure of the system. Besides, as the neighborhood level spatial data became unavailable, the neighborhood list was left blank in the final visualization. Thus, even with the advantage of being easy to transfer, the system is impractical for processing all potential modeling results at once.

## 6. Conclusions

This paper presents a geospatial platform supporting the generation and exploration of spatially detailed synthetic populations for urban planning and management. The framework of the platform centers on an R-based population synthesis module and the PostgreSQL database, with the synthetic population information presented on a front-end visualization application. In parallel, the population synthesis module was also encapsulated within an ArcGIS toolbox, which can facilitate the use by those who are familiar with the ArcGIS software. On the system level, we focus specifically on the usability, scalability, and interactivity of the population synthesis module as well as the visualization applications to make sure that urban modelers and planners would be comfortable using them.

In this study, we employed the Iterative Proportional Fitting model for the population synthesis. The conventional population synthesis process was consolidated by including a population allocation model, which is able to assign synthetic households to more spatially detailed locations. It is worth noting that the population synthesis process requires a sample of micro-level population data, which are not necessarily collected recently, and a series of aggregated distributions of population characteristics, which are supposed to be more precise and updated. The cases of Chongqing and Suzhou were introduced to illustrate the usability and transferability of the geospatial platform in sustaining population synthesis for different cities.

As the requirement for urban population exploration evolves, some emergent issues need to be addressed. First, the database structure and the visualization tools need to be more scalable and flexible to achieve a satisfactory performance for users to explore the results. For large cities with millions of buildings and tens of millions in population, the loading time for the geospatial data and synthetic population results might make the users impatient. Second, the visualization applications are mainly used for thematic mapping and charting of a synthetic population at this point. Their support for other visualization types such as temporal trends and comparison between different synthetic population realizations remains inadequate. Third, in addition to the IPF approach, other

population synthesis approaches such as entropy maximization and Bayesian network could be added to the platform as options in the future. At this stage, we are collecting feedback from researchers and proceeding to consolidate the existing system. We expect such infrastructure to enable government agencies, public and private sectors, planners, and developers to be more tightly integrated into urban modeling efforts.

**Author Contributions:** Conceptualization, J.L. (Jian Liu), J.L. (Jing Li), X.M. and Y.Z.; methodology, Y.Z.; validation, X.M., J.L. (Jian Liu) and Z.H.; formal analysis, J.L. (Jian Liu); investigation, J.L. (Jian Liu), Z.H.; resources, J.L. (Jian Liu), Z.H. and J.L. (Jing Li); data curation, J.L (Jing Li); writing— original draft preparation, J.L. (Jian Liu) and X.M.; writing—review and editing, X.M.; visualization, J.L. (Jian Liu), S.Y. and Z.H.; supervision, Z.H. and J.L. (Jing Li); project administration, S.Y., Z.H. and J.L. (Jing Li). All authors have read and agreed to the published version of the manuscript.

**Funding:** This research received no external funding.

**Data Availability Statement:** The data that support the findings of this study are available from the author, Jian Liu, upon reasonable request.

**Conflicts of Interest:** The authors declare no conflict of interest.

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
