# Peer review of "Generating and Visualizing Spatially Disaggregated Synthetic Population Using a Web-Based Geospatial Service"

_sustainability, doi:10.3390/su13031587_

Round 1
Reviewer 1 Report
The paper gives a presentation of a geospatial platform to assist urban planning and management. The text is clear and I have not much to add on the study. One point that could eventually deserve a bit further discussion is that the whole procedure rests on the availability of micro-level population data (samples). For the results to be consistent and to serve government actions, these population data need not to be as recent as possible. I would suggest adding some text underlying this point.
I would like to thank the authors for their interesting and stimulating paper. The tool/platform shall prove to be useful for urban development planners and managers.
Author Response
First and foremost, we wish to express our great appreciation to the two reviewers for their valuable comments and constructive suggestions. The paper has been revised according to all the comments and suggestions. This part provides our detailed response to each of the comments and suggestions, along with the corresponding revisions we have made.
Responses to the comments from Reviewer #1
Many thanks for your comments. Your suggestions are very valuable for improving our manuscript. We have added a discussion on the data for the population synthesis procedure in the conclusion part.
“It is worth noting that the population synthesis process requires a sample of micro-level population data, which are not necessarily to be collected recently, and a series of aggregated distributions of population characteristics, which are supposed to be more precise and updated.”
Reviewer 2 Report
The paper is well structured with clearly defined aims and methodology. There is an adequate review of previous research related to the topic. The research results are clearly presented and related to the previous research. The paper presents a significant contribution to the research field.

Author Response
First and foremost, we wish to express our great appreciation to the two reviewers for their valuable comments and constructive suggestions. The paper has been revised according to all the comments and suggestions. This part provides our detailed response to each of the comments and suggestions, along with the corresponding revisions we have made.
Responses to the comments from Reviewer #2
We appreciate your positive comments. Thank you for the time and effort. We have added a discussion on the data for the population synthesis procedure in the conclusion part.
“It is worth noting that the population synthesis process requires a sample of micro-level population data, which are not necessarily to be collected recently, and a series of aggregated distributions of population characteristics, which are supposed to be more precise and updated.”
